# CavitySpace: A Database of Potential Ligand Binding Sites in the Human Proteome

**DOI:** 10.3390/biom12070967

**Published:** 2022-07-11

**Authors:** Shiwei Wang, Haoyu Lin, Zhixian Huang, Yufeng He, Xiaobing Deng, Youjun Xu, Jianfeng Pei, Luhua Lai

**Affiliations:** 1BNLMS, College of Chemistry and Molecular Engineering, Peking University, Beijing 100871, China; wangsw93@pku.edu.cn (S.W.); vileoy@pku.edu.cn (H.L.); hzx_tom@pku.edu.cn (Z.H.); dengxb@pku.edu.cn (X.D.); 2Center for Computational Science and Engineering, Peking University, Beijing 100871, China; 3Center for Life Sciences, Academy for Advanced Interdisciplinary Studies, Peking University, Beijing 100871, China; yufenghe@stu.pku.edu.cn; 4Infinite Intelligence Pharma, Beijing 100083, China; xuyj@iipharma.cn; 5Center for Quantitative Biology, Academy for Advanced Interdisciplinary Studies, Peking University, Beijing 100871, China; 6Research Unit of Drug Design Method, Chinese Academy of Medical Sciences (2021RU014), Beijing 100871, China

**Keywords:** ligand binding sites, AlphaFold predicted structures, pocket library, CAVITY

## Abstract

Location and properties of ligand binding sites provide important information to uncover protein functions and to direct structure-based drug design approaches. However, as binding site detection depends on the three-dimensional (3D) structural data of proteins, functional analysis based on protein ligand binding sites is formidable for proteins without structural information. Recent developments in protein structure prediction and the 3D structures built by AlphaFold provide an unprecedented opportunity for analyzing ligand binding sites in human proteins. Here, we constructed the CavitySpace database, the first pocket library for all the proteins in the human proteome, using a widely-applied ligand binding site detection program CAVITY. Our analysis showed that known ligand binding sites could be well recovered. We grouped the predicted binding sites according to their similarity which can be used in protein function prediction and drug repurposing studies. Novel binding sites in highly reliable predicted structure regions provide new opportunities for drug discovery. Our CavitySpace is freely available and provides a valuable tool for drug discovery and protein function studies.

## 1. Introduction

Protein-ligand interactions govern many biological processes. The specific ligand binding site (LBS) in a protein is essential for understanding its biological function and for structure-based drug design [1]. LBSs can be directly obtained from known protein-ligand complex structures. The Protein Data Bank (PDB) [2] provides the primary source of protein three-dimensional (3D) structures experimentally resolved. However, among the more than 190,000 experimentally-determined structures by 2021, only a small number were solved with bound ligand. In order to fill the gap between structures and binding sites, computational methods predicting LBSs from protein 3D structures have been developed [3,4]. Based on the known macromolecule structures in the PDB and diverse LBS detection tools, pocket databases have been constructed, which provide functional clues for protein domains of unknown functions [5,6,7,8,9,10,11,12,13,14,15,16,17,18,19,20,21] (see Appendix A for more details about typical pocket databases). The pocketome analysis bridges the gap between the protein structures and functions. For example, Bhagavat et al. proposed an augmented pocketome (PocketDB) which consists of 249,096 pockets detected from all the proteins with known 3D structures in the PDB. Combining the pocketome database and an effective pocket-comparison algorithm, functional relationships among protein folds, binding site types and ligand types could be deduced [11]. Hedderich et al. focused on the pocketome of G-protein-coupled receptors and discovered previously untargeted allosteric sites [22].

However, currently available pocket databases are limited to known protein structures. Only about 37% of human proteins have the corresponding PDB entries [23]. Fortunately, protein structure prediction approaches have made great progress in the past several years [24,25,26]. In 2021, AlphaFold, a deep neural network-based method developed by DeepMind, has made a major breakthrough and produced protein structures with atomic accuracy even where no similar structure is known [27]. AlphaFold was then applied to build protein structure models for human proteome [28,29], which dramatically expanded the structural coverage of human proteins.

In this work, we analyzed potential ligand binding sites in the human protein structures predicted by AlphaFold and constructed a comprehensive human pocketome database, CavitySpace. CavitySpace expands the ligand binding site space from known human protein structures to predicted structures and provides a resource for human protein function study and drug design.

## 2. Materials and Methods

### 2.1. Data Collection

Figure 1 shows the flowchart for building the CavitySpace database. All data used to construct CavitySpace were obtained from public databases. The human protein structures predicted by AlphaFold were downloaded from the AlphaFold Protein Structure Database (https://alphafold.ebi.ac.uk/, accessed on 1 September 2021). Only structures for Homo sapiens were downloaded, which contain 23,391 predicted structures of 20,504 sequence entries. The inequality is due to the fact that 208 sequence entries were divided into several fragments before structure prediction because of the extremely long sequence length.

About 37% of human proteins have corresponding PDB entries. It is very useful to compare the known and predicted structures. Therefore, we queried the UniProtKB database (https://www.uniprot.org/, accessed on 1 October 2021) with the UniProt ID of each sequence to retrieve UniProt entries with known PDB structures and obtained a total of 7245 UniProt records. For each UniProt entry, the structure with the best resolution was selected as a representative structure. Sometimes, several PDB structures for one protein cover different domains of the same protein. In these cases, we selected representative structures for each domain of the sequence to cover the whole protein sequence as long as possible. In addition, all PDB structures not resolved by X-ray crystallography or with resolution larger than 3.5 Å were excluded. Finally, we screened 6956 PDB structures of 5725 UniProt entries, forming the known human PDB structure dataset (hrefPDB). All the structure files were downloaded from RCSB PDB (https://www.rcsb.org/, accessed on 8 November 2021). Because each AlphaFold structure has only one single chain, we extracted one chain from each of the known structures to maintain consistency.

### 2.2. Cavity Detection 

We applied the CAVITY tool developed by our lab to detect all the potential cavities on protein surfaces [30]. CAVITY has been shown to be effective in detecting cavities and assessing pockets’ druggability on the NRDLD data set [31]. For all the 23,391 AlphaFold structures, CAVITY successfully processed 18,820 (80.5%) structures. The remaining 4571 structures, for which CAVITY could not finish the job within a reasonable time, were mainly complicated structures with relatively long protein sequences and many irregular loops. For the hrefPDB dataset containing resolved PDB structures, all the 6956 structures can be processed by CAVITY and 86.9% (6043) of them have at least one cavity.

## 3. Results

### 3.1. Cavity Library for AlphaFold Structures

The cavity detection procedure found 237,872 cavities for the 18,672 AlphaFold structures. The druggability of each cavity was labelled as strong, medium or weak by CAVITY. Among the AlphaFold cavities, 16.3% (38,709) were predicted as strong druggable cavities (Figure 2). However, for AlphaFold cavities, we need to further consider the structure accuracy, because inaccurate structures may produce fake cavities. In AlphaFold structures, the predicted Local Distance Difference Test (pLDDT) was given for each residue to measure the local accuracy [27]. This confidence measure is a scale from 0 to 100. Structure regions with pLDDT > 90 are expected to be highly confident and can be used to characterize binding sites. Based on the pLDDT scores, we defined the ratio of the number of high confident residues (pLDDT > 90) to the total number of residues in the cavity as an *Index* to evaluate the reliability of the cavity structure. Among all the AlphaFold cavities, about 40% are located in protein structure regions with relatively high confidence (*Index* > 0.6), while only about 26% of strong cavities are constructed by reliable structures (Figure 3). 

As previously mentioned, about 37% of human proteins have corresponding PDB entries, so we split all the AlphaFold cavities into two subsets: one subset contains all cavities from proteins that have known 3D structures in our hrefPDB dataset (Subset_1), and the other subset contains the remaining cavities (Subset_2). Figure 3 shows the *Index* distributions of these two subsets. The median *Index* of Subset_1 is higher than that of Subset_2, implying that AlphaFold gives more accurate structure predictions for proteins with known 3D structures. However, for cavities with strong druggability, the *Index* values significantly decreased, indicating that the AlphaFold predicted protein structures may contain false-positive druggable pockets. Therefore, we include the *Index* scores for each cavity in our library to help users pick reliable druggable pockets.

### 3.2. The Quality of Cavity Detection over AlphaFold Structures 

One important question is how different are the hrefPDB structures from the AlphaFold predicted structures for cavity detection. We further performed cavity detection for all the hrefPDB structures and obtained 50,417 PDB cavities. To make a fair comparison, we extracted the subset of AlphaFold structures sharing the same UniProt IDs to the hrefPDB structures and then collected 64,897 corresponding AlphaFold cavities. Theoretically, the cavities detected from hrefPDB structures should also be detected from AlphaFold predicted structures if the AlphaFold predicted structures are as reliable as the experimentally resolved structures. Therefore, we defined a score of Tc = A∩B/A∪B to measure the overlap of two cavities, where A and B are the residue sets of two cavities. We defined two cavities as the same pocket if their Tc score is larger than a given threshold. Figure 4a shows the recall rate of hrefPDB cavities in AlphaFold cavities at different thresholds. With a threshold of 0.5, only 22.8% of hrefPDB cavities can be detected from AlphaFold predicted structures, indicating non-negligible differences between the predicted structures and the real structures. This can be attributed to the following reasons. First, AlphaFold structures are always full-sequence structures, while part of the hrefPDB structures are not full-sequence structures and have missing residues, which affect the detection of cavities. Second, cavity detection results are influenced by protein conformation, which might be diverse under different ensemble states, binding states and experimental conditions. However, each AlphaFold structure from AlphaFold database gives only one single conformation, which may be different from the conformation of hrefPDB structures. For example, the AlphaFold structure of alpha-actinin-2 (UniProt ID: P35609) is different from its hrefPDB structure (PDB entry: 4D1E) [32] (see Appendix A for details). Third, the AlphaFold program can produce relatively accurate secondary structures, but the relative position of different secondary structures and different domains might not be very accurate, not to mention the inaccurate position of flexible loops. These inaccurate predictions can significantly influence the cavity detection process.

In the above analysis, we checked if hrefPDB cavities can be detected from AlphaFold predicted structures. However, the hrefPDB cavities may change shape upon ligand binding. In order to obtain more accurate results, we further checked if experimentally confirmed ligand sites in hrefPDB structures can be correctly identified by CAVITY from AlphaFold structures. From the hrefPDB dataset, we only kept the structures with bound ligands and defined true ligand sites as residues within 4 Å around bound ligands. This gave 2439 true ligand binding sites as a test set. If we define that a true ligand binding site is successfully detected when 50% of true binding site residues are included in one of the CAVITY detected binding sites, 81% and 80% of true binding sites can be recovered from hrefPDB and AlphaFold structures, respectively (Figure 4b). Such results demonstrate that our CAVITY program can successfully discover most of the true binding sites from protein structures and for the true binding site detection task, there is no significant difference between hrefPDB structures and AlphaFold predicted structures.

### 3.3. Applications of the Cavity Library

For the 63.6% of the AlphaFold predicted human protein structures with no experimental structure information, CAVITY detected 145,444 cavities and 17.4% of them are strong druggable cavities. As similar binding sites may bind the same or similar ligands and have similar functions, we used PocketMatch [33] to compare binding sites and the PMSmax score to evaluate the overall pocket similarity (see the Appendix A for details). To obtain meaningful results, we only analyzed the 60,913 high-quality cavities, each of which contains at least 80% residues with pLDDT > 90. These high-quality cavities, together with 50,417 hrefPDB cavities, were used to perform an all-to-all pocket comparison. We then clustered the 111,330 cavities with the Butina algorithm [34] (see Appendix A). With the threshold of PMSmax 0.6, 11,213 cavities did not have any similar cavities, which may be novel ligand binding sites. The other 100,117 cavities were grouped into 8015 clusters and 538 of them contain more than 10 members. The clusters that contain known ligand binding sites can be used to study the function of proteins that contain similar cavities or to find new targets for a known ligand. For example, the crystal structure of human cysteinyl leukotriene receptor one binding with its antagonist zafirlukast, an FDA approved drug for asthma treatment, has been solved [35]. We selected seven top-scoring AlphaFold cavities of proteins without known PDB structures from the corresponding cavity cluster to which the zafirlukast binding site belongs (see Supplementary Results for details). Docking study with Autodock Vina 1.2 [36] showed that zafirlukast can potentially bind to these cavities with high binding affinity (Appendix A).

### 3.4. The Webserver

We developed the CavitySpace webserver for public usage. Users can conveniently query the database with protein name, UniProt ID or gene name and obtain the cavity details for each structure visually. All data in the cavity library can be downloaded from the CavitySpace webserver, including the strong druggable cavities, the cavity clustering results, etc. Each cavity is represented by four files. The “cavity” file contains all the protein atoms that form the cavity. The “surface” file contains all the grid points on protein surface which form the cavity. The “vacant” file contains all the grid points in the cavity. Besides, the bounds, the surface area and the volume of each cavity can be found in the “surface” file and the “vacant” file, respectively. The “box” file contains the coordinates of the cavity center and the size of the docking box, which can be directly used for docking calculation by users. The webserver is freely available at http://www.pkumdl.cn:8000/cavityspace/ (accessed on 8 November 2021).

## 4. Discussion

The shape and physicochemical properties of ligand binding sites provide useful information for protein function deduction. The large number of protein structures predicted by AlphaFold provides an unprecedented chance to analyze binding sites in previously unexplored structure space. The CavitySpace database can be used for various purposes, including identifying new druggable protein targets for drug design, predicting protein function based on pocket analysis, searching for new binding sites for known drugs for drug repurpose study, etc. For example, the cavities in CavitySpace provide a structural basis to screen potential active ligands. Molecular docking is commonly used to predict potential binding pose and affinity of a ligand in a binding site. For a given cavity, we can dock all the molecules in a compound library to the cavity to screen potential active ligands. In reverse, for a known drug or a compound of interest, we can also dock the compound to all the cavities in CavitySpace to find novel potential binding sites. Based on the hypothesis that similar binding sites may bind similar ligands, we can also calculate the cavity similarity between cavities and screen the similar cavities of a specific cavity with a known bound ligand. This ligand is likely to bind with these similar cavities.

At the same time, care should be taken when using these data. This can be attributed to several possible reasons. First, the AlphaFold method is currently restricted to single-chain structure prediction. In real cases, many proteins need to form oligomers to be functional. New methods, such as the AlphaFold-Multimer [37] have been developed to address this problem. CavitySpace will be updated once the structure data of the protein oligomers is released. Second, AlphaFold provides the state-of-the-art structure predictions in general, but the accuracy varies between different domains and different structures. Pockets located on low confident domains are obviously unreliable to be treated as a real pocket. We recommend that based on the CavitySpace results, users further analyze the potential binding sites with more accurate structures after carefully considering inter-domain orientations and oligomeric states using our CavityPlus webserver [38] or other cavity detection tools.

The development of AlphaFold and related methods have markedly advanced the protein 3D structure prediction field. However, for most drug discovery tasks, in addition to the overall correct protein structure, reliable binding site sub-structure is necessary. Based on the CavitySpace database, we found that the single-chain structures, the less reliable relative position between different secondary structures or domains, and the flexible loops of the currently available AlphaFold models are the major obstacles for highly accurate prediction of ligand binding sites. Care should be taken to check the reliability of the structures that form the binding sites before further analysis. More efforts are needed to solve the above problems in protein structure prediction. Learning the binding site structures in addition to the overall protein structures may provide a useful strategy to increase the accuracy of predicted protein structures.

## 5. Conclusions

We have developed CavitySpace, a comprehensive human protein pocketome based on the AlphaFold predicted human protein structures. CavitySpace not only captures known ligand binding sites, but also identifies novel ligand binding sites that can be used to predict protein functions or to virtually screen new binding compounds in drug discovery. CavitySpace can also be used to identify new targets for known drugs in drug repurposing or side effect studies. The cavity library can be easily applied to docking-based virtual screening or reverse virtual screening to screen active ligands or identify new targets for a given drug or natural product. Furthermore, the cavity library increases the diversity of ligand binding sites, which can be applied to the pre-training stage of AI-based drug discovery models. Though the current version of CavitySpace only contains human proteins, it can be easily extended to include more protein structures from other species in the future.

## Figures and Tables

**Figure 1 biomolecules-12-00967-f001:**
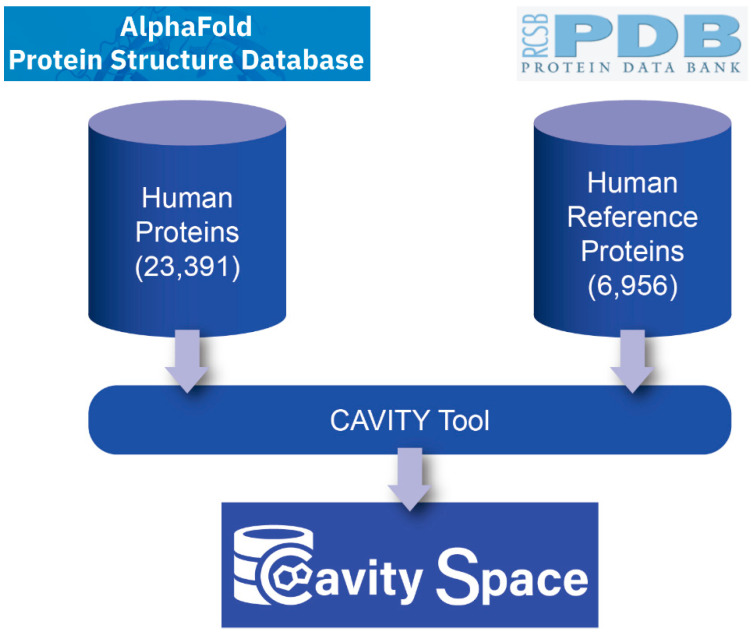
The flowchart for building CavitySpace.

**Figure 2 biomolecules-12-00967-f002:**
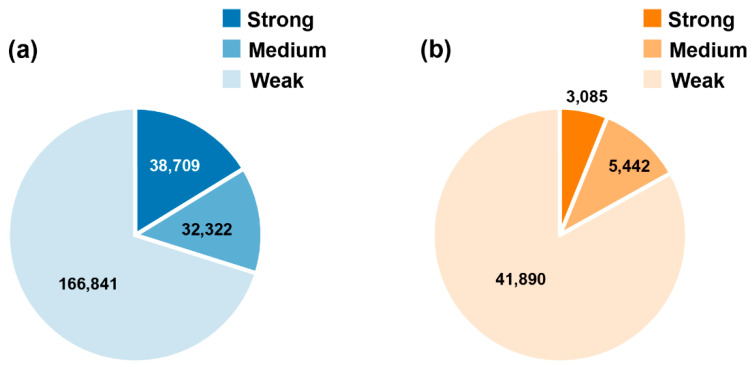
The druggability distribution of AlphaFold cavities (**a**) and hrefPDB cavities (**b**).

**Figure 3 biomolecules-12-00967-f003:**
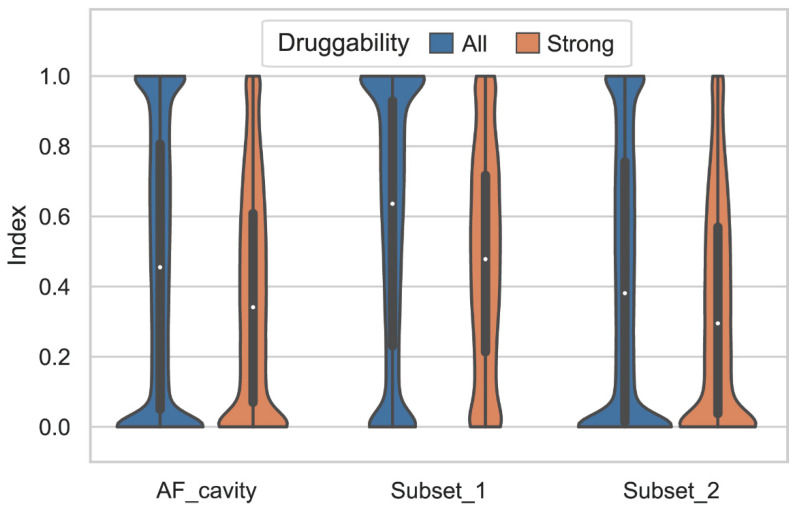
The distributions of *Index* that is the proportion of cavity residues with high confidence (pLDDT > 90). AF_cavity: cavities detected from AlphaFold predicted protein structures; Subset_1: cavities from proteins that have known 3D structures in our hrefPDB dataset; Subset_2: the remaining cavities after removing Subset_1 from AF_cavity. The distributions of all cavities (blue) and cavities with strong druggability (orange) in each dataset are displayed respectively.

**Figure 4 biomolecules-12-00967-f004:**
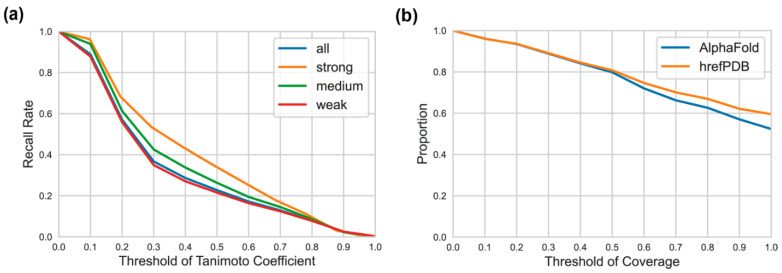
(**a**) The recall rate of hrefPDB cavities at different Tc thresholds. (**b**) The proportion of true ligand binding sites recovery under different thresholds of residue coverage for both hrefPDB structures and AlphaFold structures. The threshold of residue coverage means how many true binding site residues were included in one of the CAVITY detected binding sites.

## Data Availability

CavitySpace can be accessed at: http://www.pkumdl.cn:8000/cavityspace/ (accessed on 8 November 2021). AlphaFold protein structures are obtained from https://alphafold.ebi.ac.uk/ (accessed on 1 September 2021). UniProt information is available at https://www.uniprot.org/ (accessed on 1 October 2021). PDB structures and sequence alignment data were downloaded from https://www.rcsb.org/ (accessed on 8 November 2021). The CAVITY program is available at http://www.pkumdl.cn:8000/cavityplus/index.php (accessed on 1 September 2021).

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
