# Peer review of "CavitySpace: A Database of Potential Ligand Binding Sites in the Human Proteome"

_biomolecules, 2022, doi:10.3390/biom12070967_

Round 1
Reviewer 1 Report
The paper presents technique for cavity identification in proteins. The method applied is based on the distance restrictions. As such belongs to stochastics techniques.
My main objection is the lack of Conclusions.
It looks like the Authors by themselves do not know what is the added value of their paper.
I strongly expect the Conclusion part in the final form of the paper. The scientific meaning shall be presented there. The preparation of data base makes this paper of low level in the cathegory of discovery.
Author Response
We thank the reviewer for the comments and suggestions. We have added a Conclusion section and updated the Introduction in the manuscript and emphasized the scientific significance of our work.
Structure-based drug design depends on the 3D structural information of proteins. In this work, we have developed CavitySpace, a comprehensive human protein pocketome based on the AlphaFold predicted human protein structures. CavitySpace not only captures known ligand binding sites, but also identifies novel ligand binding sites that can be used to predict protein functions or to virtually screen new binding compounds in drug discovery. CavitySpace can also be used to identify new targets for known drugs in drug repurposing or side effect studies. The cavity library can be easily applied to docking-based virtual screening or reverse virtual screening to screen active ligands or identify new targets for a given drug or natural products. Furthermore, the cavity library increases the diversity of ligand binding sites, which can be applied to the pre-training stage of AI-based drug discovery models. Though the current version of CavitySpace only contains human proteins, it can be easily extended to include more protein structures from other species in the future.
Reviewer 2 Report
The authors have developed an original database of cavities in model structures of human proteins. Such a database is valuable for testing the effect of drugs and predicting possible side effects. The article is written in understandable language.
Some points remained incomprehensible to me. I would like these points to be covered in the text of the article.
1. What were the criteria for identifying cavities in a protein?
2. What criteria were used to divide cavities into strong, medium and weak categories?
3. How do the found cavities correlate with ligands?
4. What do the result files contain for each cavity (surface, cavity, vacant)?
5. How can somebody use your database in automatic mode for virtual screening, for example? In the Vina AutoDock program, for example, each site is characterized by its center in molecular coordinates and cell dimensions (x, y, z). If you characterized your found cavities with such information, then your database could be used and cited by scientists studying possible target proteins for drugs.
Round 2
Reviewer 1 Report
Authors followed my comments.
The paper can be published in the current form.